# Real-time prediction of rain-triggered lahars: incorporating seasonality and catchment recovery

Robbie Jones[a*], Vern Manville[a], Jeff Peakall[a], Melanie Froude[bc], Henry Odbert[de]

[a]School of Earth and Environment, University of Leeds, Leeds. LS2 9JT, United Kingdom
[b]School of Environmental Sciences, University of East Anglia, Norwich, NR4 7TJ, United Kingdom
[c]Department of Geography, University of Sheffield, 9 Northumberland Road, Sheffield, S10, UK
[d]School of Earth Sciences, University of Bristol, Wills Memorial Building, Queens Road, Bristol BS8 1RJ, United Kingdom
[e]Met Office, FitzRoy Road, Exeter, Devon, EX1 3PB, United Kingdom

[*]*Correspondence to:* Robbie Jones (robbie_j_jones@outlook.com)

**Abstract.** Rain-triggered lahars are a significant secondary hydrological and geomorphic hazard at volcanoes where unconsolidated pyroclastic material produced by explosive eruptions is exposed to intense rainfall, often occurring for years to decades after the initial eruptive activity. Previous studies have shown that secondary lahar initiation is a function of rainfall parameters, source material characteristics and time since eruptive activity. In this study, probabilistic rain-triggered lahar forecasting models are developed using the lahar occurrence and rainfall record of the Belham River Valley at Soufrière Hills Volcano, Montserrat collected between April 2010 and April 2012. In addition to the use of peak rainfall intensity as a base forecasting parameter, considerations for the effects of rainfall seasonality and catchment evolution upon the initiation of rain-triggered lahars and the predictability of lahar generation are also incorporated into these models. Lahar probability increases with peak one-hour rainfall intensity throughout the two-year dataset, and is higher under given rainfall conditions in year one than year two. The probability of lahars is also enhanced during the wet season, when large-scale synoptic weather systems (including tropical cyclones) are more common and antecedent rainfall and thus levels of deposit saturation are typically increased. The incorporation of antecedent conditions and catchment evolution into logistic regression-based rain-triggered lahar probability estimation models is shown to enhance model performance and displays the potential for successful real-time prediction of lahars, even in areas featuring strongly seasonal climates and temporal catchment recovery.

## 1 Introduction

Lahars are rapidly flowing mixtures of rock debris and water (other than normal streamflow) from a volcano and represent a significant hazard due to their energetic nature and mobility (Smith and Fritz, 1989). Globally, 17% of historical volcano-related fatalities have occurred due to lahars (Auker et al., 2013); with decadal-scale hazards being created by some large eruptions (Major et al., 2000). Secondary, post-eruption lahars are dominantly the result of rainfall on unconsolidated pyroclastic deposits, which are typically remobilised by rilling due to Hortonian overland flow (Segerstrom, 1950; Waldron, 1967), undercutting and lateral bank collapse and headward erosion (Pierson, 1992); or by shallow landsliding of saturated tephra layers above basal décollement surfaces (Iverson, 2000; Manville et al., 2000).

At present, rain-triggered lahar hazard identification is predominantly based on observations as well as ground-based flow detection systems such as Acoustic Flow Monitors (AFMs) or trip-wires at locations where such

resources are available (e.g. Marcial et al., 1996; Lavigne et al., 2000). Previous studies featuring post-lahar
analysis of flow observations and rainfall records at a range of volcanoes have displayed a power-law relationship
indicating that lahar initiation occurs along a continuum from short duration, high intensity rainfall events to long
duration, low-intensity events (e.g. Rodolfo and Arguden, 1991; Capra et al., 2010; Jones et al., 2015). Enhancing
the use of local telemetered rainfall gauge networks within lahar hazard monitoring and assessment has the
potential to increase the number of available mitigation tools whilst avoiding the lag-time between flow initiation
and flow detection inherent in ground-based detection and observation. Globally, such pre-emptive prediction and
forecasting of rain-triggered lahars based on telemetered rainfall data is lacking, although initial application of
real-time rainfall data for lahar prediction has demonstrated increased lahar warning times compared with ground-
based flow detection (Jones et al., 2015).
The initiation of rain-triggered lahars is dependent on the characteristics of rainfall, pyroclastic deposits and
topography, indicating that both the climatic regime of lahar-prone regions and the hydrogeomorphic response of
drainage basins to eruptive activity are important considerations in rain-triggered lahar research (Pierson and
Major, 2014). Regions of high rainfall seasonality are predominantly distributed in the tropics and sub-tropics
either side of the equator (Wang et al., 2010); whilst approximately 46% of active volcanoes are identified as
being located in the humid tropics (Rodolfo and Arguden, 1991). Despite this geographic coincidence and the
importance of climatic rainfall regimes on storm intensities, durations and antecedent conditions (all significant
factors in lahar initiation: Pierson and Major, (2014)), the impact of seasonal rainfall on rain-triggered lahar
initiation has not previously been explicitly considered within the development of rain-triggered lahar hazard
assessment tools.
Following a discrete volcanic eruption, sediment yields in impacted fluvial systems are amongst the highest
recorded globally, but decline exponentially (Major et al., 2000), which is consistent with other examples of
disturbed earth systems (Graf, 1977). Mechanisms include a reduction in available particulate material, vegetation
recovery, fragmentation of runoff-enhancing surface crusts, exposure of more permeable substrates and the
stabilisation of rill networks (Leavesley et al., 1989; Schumm and Rea, 1995; Major et al., 2000; Major and
Yamakoshi, 2005). Conversely, at locations featuring recurrent or persistent volcanic activity, the magnitude of
the lahar hazard remains relatively constant with time due to the regular supply of new material (Thouret et al.,
2014). As a result, temporal catchment development is another factor influencing lahar frequency and magnitude
through time, and should also be considered within the development of rain-triggered lahar hazard assessment
tools.
This study uses probabilistic and diagnostic methods, including binary logistic regression and Receiver Operating
Characteristic (ROC) analysis, to develop real-time rainfall-based lahar forecasting tools which account for the
impacts of seasonal rainfall and catchment recovery on lahar occurrence in the Belham Valley, Montserrat. Such
hazard assessment tools have the potential to be utilised both as a stand-alone tool where ground-based detection
equipment is unavailable, and in conjunction with instrumental monitoring techniques to increase lahar warning
times.
**2 Soufrière Hills Volcano, Montserrat**
Soufrière Hills Volcano (SHV, Montserrat, Lesser Antilles, 16.72°N, 62.18°W) lies on the northern edge of the
Inter-Tropical Convergence Zone in the eastern Caribbean and has a strongly seasonal climate. Rainfall-producing

weather systems affecting the island fall into two broad categories; large-scale synoptic (>100 km across) systems and local mesoscale (<100 km across) systems (Froude, 2015). Both can produce high intensity precipitation, but large-scale events can potentially be forecast days in advance whereas this timescale reduces to hours for local weather systems (Barclay et al., 2006).

The andesitic dome-forming eruption of SHV began in July 1995 and has featured several phases of activity consisting of dome growth, dome collapse and Vulcanian explosions as well as pauses in magma extrusion (Bonadonna et al., 2002; Komorowski et al., 2010; Stinton et al., 2014). Pyroclastic density currents (PDCs) have deposited fine-grained ash- and pumice-rich and coarser-grained blocky deposits around the volcano (Cole et al., 2002; Stinton et al., 2014), supplemented by tephra deposits from short-lived Vulcanian explosions and associated fountain-collapse flows and surges (Komorowski et al., 2010). Prevailing winds often distribute ash from weak plumes to the West, but larger plumes can also deposit to the North, East and South (Bonadonna et al., 2002). This intermittent eruptive activity has triggered a complex sedimentological response in drainages surrounding the volcano since 1995 (Barclay et al., 2006, 2007; Alexander et al., 2010; Froude, 2015).

## 3 The Belham Catchment

Data from the Belham Valley, Montserrat (Fig. 1) were used to examine the influence of rainfall seasonality and catchment evolution on the occurrence of rain-triggered lahars between April 2010 and April 2012 (Fig. 2). Lahars have persisted in the valley since the onset of eruptive activity in 1995 and detailed observations of lahars in the Belham Valley have indicated that they are dominantly Newtonian and fully turbulent (Barclay et al., 2007; Alexander et al., 2010; Froude et al., 2017). Lahars have damaged infrastructure, including burying the Belham Bridge in 1998, resulting in the river bed being used as the primary transportation link between the "Safe Zone" and the "Daytime Entry Zone" (Barclay et al., 2007; Alexander et al., 2010).

The Belham Catchment had a pre-1995 surface area of c. 13.7 $km^2$, increasing to c. 14.8 $km^2$ early in the eruptive episode due to capture of a portion of Gage's fan (Froude, 2015). During eruptive episodes tephra fall and pyroclastic density current (PDC) deposits accumulate in the upper catchment. The destruction and burial of vegetation in the Belham Valley reduces the infiltration and interception of precipitation, and in combination with a reduction in surface roughness enhances run-off and erosion rates and promotes rain-triggered lahar generation (Barclay et al., 2007; Alexander et al., 2010; Froude, 2015). Prior to the onset of eruptive activity, 62% of the Belham Catchment was densely vegetated with Dry Forest (29%), Mesic Forest (48%) and Wet Forest (13%), with dry forest subsequently identified as the dominant species found on re-vegetating pyroclastic deposits (Froude, 2015). Previous studies in the Belham Valley have not identified evidence of hydrophobicity, such as previously identified at Colima by Capra et al. (2010). Aggradation and sedimentation in the upper catchment during periods of eruptive activity are counter-balanced during periods of quiescence by channel development and stabilisation, exposure of more permeable substrates, vegetation recovery and a reduction in available sediment (Froude, 2015). The data period used here coincides with a lack of substantial eruptive activity at SHV following the 11[th] of February 2010 dome collapse at the end of "Phase 5", which deposited stacked lobes of pumiceous PDC deposits up to 5.7 km from source in the Belham Valley (Stinton et al., 2014). This period of eruptive quiescence indicates that this study focuses on a time of channel development and stabilisation within the upper catchment of the Belham Valley.

## 4 Methods

The record used in this study (Fig. 2) comprises 0.1 mm resolution hourly precipitation data recorded at the MVO Helipad Gauge between February 2010 and February 2011, the St George's Hill gauge between March 2011 and May 2011, and the maximum of the St George's Hill and Windy Hill gauges (Fig. 1) between May 2011 and February 2012. While a continuous record from rain gauges with a better spatial distribution and density would be ideal to minimise differences in catch efficiencies and to capture local variations in convective and orographic rainfall, operating a fully functioning rain gauge network is technically challenging and generally a low priority during a volcanic crisis. The lahar database (Fig. 2) is compiled from inspection of seismic records and visual observations and lahars are categorised based on magnitude (small, medium, large). These categories were assessed using visual inspection of the degree of channel inundation and flow depth (where possible); in addition to the assessment of the duration and amplitude of seismic signals. Seismic signals of lahars show continuous readings in the 2-5 Hz and peak at approximately 30 Hz. The highest recorded amplitudes are associated with the greatest discharges and sediment loads in observed lahars. Lahar signals were cross referenced to visual observations and carefully excluded from signals associated with primary volcanic activity and other seismic noise (such as construction vehicles).

Within this study a designated minimum inter-event dry period of six hours is utilised, meaning that in common with several previous soil erosion studies a dry interval of six hours is needed to define the end of a single rainfall event (Wischmeier and Smith 1978; Todisco, 2014). Figure 3 shows six examples of rainfall events (or series of consecutive rainfall events) which resulted in the observation or detection of lahars in the Belham Valley, clearly displaying the lag time between the recording of rainfall (cumulative- and real-time progression of One Hour Peak Rainfall Intensity: 1hr PRI) and the observation/detection of lahars. 1hrPRI has been identified as an effective parameter in lahar initiation threshold assessment during previous analysis (Jones et al., 2015). Division of the dataset into six-month moving windows, with staggered one-month start dates, facilitates the illustration of the seasonal variation in both the number of rainfall events exceeding 1hrPRI thresholds and the occurrence (and estimated magnitude) of lahars (Fig. 4).

This study uses binary logistic regression to develop lahar probability estimation models based on the 1hrPRI of a rainfall event, whilst also examining the impacts of incorporating considerations for seasonal and temporal effects within these models. Binary logistic regression is a statistical method that estimates the probability of a dichotomous outcome (the occurrence or non-occurrence of lahars in this case) using one or more independent variables (Hosmer Jr et al., 2013). Model performance is assessed using both the model chi-square test and Receiver Operating Characteristic (ROC) analysis (Fawcett, 2006). ROC analysis (Appendix 1) plots the true positive rate against the false positive rate as a threshold (estimated lahar probability in this instance) is varied in order to assess how effectively the parameter discriminates between lahar and non-lahar producing rainfall events. The area under the ROC curve (AUC) is a measure of the ability of a tool to distinguish between the two outcomes, and varies between 0.5 (no predictive ability, i.e. number of true positives equals number of false positives, or no better than guessing) and 1.0 (perfect predictive ability, i.e. 100% true positives and no false positives).

## 5 Results

The six-month window between April and October is identified as the peak wet season in this study, with 1721 mm of recorded rainfall in the 2010 peak wet season (WS1) and 1455 mm in the 2011 peak wet season (WS2). The 2010/11 peak dry season (DS1) featured approximately 750 mm of rainfall, whilst 1076 mm of rainfall was recorded in the 2011/12 peak dry season (DS2). Mean WS1 and WS2 1hrPRIs are 5.2 mm hr$^{-1}$ and 5.0 mm hr$^{-1}$ respectively, whilst mean dry season 1hrPRIs are 2.2 mm hr$^{-1}$ (DS1) and 3.3 mm hr$^{-1}$ (DS2).

There is significant (p <0.01) correlation between recorded rainfall on timescales of 1-168 hours and lahar occurrence. When lahars are categorised by estimated magnitude, large lahars are strongly correlated with longer-duration (>24 hours) rainfall events, produced by the passage of synoptic weather systems. Between April 2010 and April 2012 large flows were directly attributed to several named tropical cyclones (Fig. 2). In contrast, smaller lahars display increased correlation with the passage of short-duration (<24 hours) rainfall events, more commonly associated with mesoscale weather systems.

### 5.1 Probabilistic rain-triggered lahar analysis

The correlation between recorded peak rainfall intensity and the subsequent occurrence of lahars (Fig. 3) provides the platform for probabilistic analysis of lahar occurrence based on the 1hrPRI of a rainfall event. Results show that lahar probability increases with greater 1hrPRI throughout the two-year study period. For example, of the 18 rainfall events which exceeded a 1hrPRI of 25 mm hr$^{-1}$, 15 were associated with the triggering of lahars, and all of the rainfall events exceeding a 1hrPRI of 34 mm hr$^{-1}$ triggered lahars. Additionally, higher lahar probabilities are observed in year 1 than year 2 for a specified 1hrPRI (Fig. 5), and empirically-derived lahar probabilities for rainfall events featuring a given minimum 1hrPRI also fluctuate seasonally during the study period (Fig. 6). These 1hrPRI exceedance-based lahar probabilities (Fig. 6) are initially stable during the 6-month windows focused on WS1 before decreasing during DS1, increasing during WS2 and once again decreasing into DS2. This indicates that more intense rainfall is required to trigger lahars in the dry season than in the wet season. Throughout the two-year study period increased 1hrPRI correlates with increased lahar probability, displaying its effectiveness as a potential first-order lahar forecasting parameter.

In addition to seasonal fluctuations in relative lahar probability, there is an overall decline in relative lahar probabilities across the two-year study period (Figs. 5 & 6). The relationship between 1hrPRI and lahar occurrence as well as the combination of seasonal fluctuation and temporal decline in lahar probability displayed in Figure 6 are examined further using binary logistic regression. In this instance the occurrence or non-occurrence of lahars (of any magnitude) is used as the dichotomous dependent variable and initially the 1hrPRI of a rainfall event is the singular independent variable. Figure 7 displays logistic regression-based lahar probability estimation models generated by this single-variable approach using four sub-datasets; *Year 1*, *Year 2*, *Wet Seasons* and *Dry Seasons*. Within each of these four models the model chi-square test indicated statistically significant lahar prediction ability (p <0.01). Figure 7 displays higher estimated lahar probabilities at identical 1hrPRI values for Year 1 relative to Year 2 and Wet Seasons relative to Dry Seasons.

The potential benefit of incorporating considerations for seasonal and temporal effects within lahar forecasting models was investigated using further binary logistic regression. This approach selected alternate chronological rainfall events (minimum total rainfall ≥8 mm) from the two-year dataset, creating a model formulation dataset consisting of 74 rainfall events, of which 25 produced lahars. Lahar forecasting models were created from this

model formulation dataset using binary logistic regression, and the remaining 73 rainfall events, of which 20
produced lahars, were retained for the assessment of the performance of the lahar forecasting models. Proxies for
seasonal effects (antecedent rainfall on timescales of 1-90 days) and catchment recovery (long-term cumulative
rainfall and days since significant eruptive activity) were tested in combination with 1hrPRI. The minimum event
rainfall threshold of 8 mm (under which only two lahars occurred during the two-year dataset) was implemented
for logistic regression and subsequent forecasting assessment in order to increase the balance between lahar and
non-lahar outcomes and thus reduce skewed predicted probability.
Three-day antecedent rainfall displayed the biggest influence of the tested antecedent rainfall timescales upon the
effectiveness of lahar forecasts, while total cumulative rainfall since significant eruptive activity (i.e. the end of
Phase 5) best captured temporal catchment development effects. Therefore, the optimal lahar forecasting model
developed from the model formulation dataset utilises 3-day antecedent rainfall and long-term cumulative rainfall
alongside the first-order lahar forecasting parameter of 1hrPRI. A 3-day antecedent period was also used by Capra
et al. (2010) at Colima, whereas a 7-day period was used in Indonesia (Lavigne et al., 2000; Lavigne and Suwa,
2004) where rainfall is higher and evaporation rates lower, and a 24-hour period was used at Mount Yakedake
(Okano et al., 2012). The optimal antecedent rainfall timescale is a function of local climate (Capra et al., 2010)
and the grain-size distribution of the pyroclastic deposits (Rodolfo and Arguden, 1991).
The reverse stepwise logistic regression method (Hosmer Jr et al., 2013), which involves the deletion of variables
whose removal from the model results in a statistically insignificant deterioration of model performance, retained
these three independent variables (1hrPRI, 3-day antecedent rainfall and total cumulative rainfall since significant
eruptive activity). This model composition increased correct classification of rainfall event outcomes in the model
formulation dataset from a null model value of 66% (when all events in the database are predicted to not trigger
lahars) to 80% when using our explanatory variables, with model chi-square tests again indicating significant
prediction ability ($p<0.01$). Model variables ($X_i$) and output regression coefficients ($\beta_i$) are used to construct lahar
probability estimation equations by conversion of the logistic regression logit model (Eq. 1) in terms of
probability.
(1)      $logit(p) = \beta_0 + \beta_1 X_1 + \beta_2 X_2 + \cdots + \beta_n X_n$
Eq. 2 displays the application of this to the multi-variable model, featuring the probability of lahar occurrence *(p)*,
1hrPRI *($R_i$)*, three-day antecedent rainfall *($A_3$)* and cumulative rainfall since significant eruptive activity (*C*).
(2)      $p = \dfrac{1}{1 + e^{-(-2.10 + 0.133R_i + 0.018A_3 - 0.215C)}}$
Eq.3 displays the lahar probability estimation model produced by the same dataset using only 1hrPRI as an
independent variable.
(3)      $p = \dfrac{1}{1 + e^{-(-2.33 + 0.133R_i)}}$
Application of Eqs. 2 & 3 to the 73 rainfall events in the forecasting assessment dataset produced two sets of
model-derived lahar probability estimates. The lahar forecasting performance of the two models was then assessed
relative to the actual outcomes (lahar or no lahar) of the rainfall events using ROC analysis. The multiple-variable
lahar probability estimation model shown in Eq. 2 produced an AUC of 0.83 ($p<0.01$), whilst the single variable
model shown in Eq. 3 produced an AUC of 0.79 ($p<0.01$) (Fig. 7B). The AUC produced by Eq. 2 increases to
0.93 if the 8 mm event threshold is removed and the multi-variable model is applied to all 508 rainfall events that
were not used in model formulation (AUC given by Eq. 3 increases to 0.89 for equivalent parameters).

**6 Discussion**

Analysis of the Belham Valley lahar occurrence and rainfall record over a two-year period indicates that lahar probability and magnitude is a function of: (i) temporal catchment evolution towards more stable conditions – lahars are harder to trigger with time; and (ii) seasonal variations in rainfall – lahars are more common in the wet season both in terms of frequency and probability relative to 1hrPRI.

The multi-year temporal trend is attributed to a declining supply of easily erodible pyroclastic material in the upper catchment, coupled with stabilisation of channel networks, vegetation re-growth, and increased infiltration as identified in several previous studies of lahar-prone regions following eruptive activity (e.g. Leavesley et al., 1989; Schumm and Rea, 1995; Major et al., 2000; Major and Yamakoshi, 2005). However, direct comparisons with other lahar-prone settings is not possible as differences in methodologies mean that common metrics such as sediment yield were not determined. The occurrence of several large rainfall events following Phase 5 of the eruption (Fig. 2) triggered a number of high-magnitude lahars within the Belham Valley, enhancing temporal channel development within the catchment and resulting in the widespread erosion and downstream transportation of pyroclastic material (Froude, 2015). Rapid re-vegetation during periods of eruptive quiescence has also been identified in the catchment (Froude, 2015), a process which increases infiltration, interception, evapotranspiration and surface roughness; reducing post-eruption runoff rates (Yamakoshi and Suwa, 2000; Ogawa et al., 2007; Alexander et al., 2010). Temporal increase in infiltration rates in the Belham Valley is also attributed to the exposure of more permeable substrates following the erosion of fine-grained surface tephra layers (Froude, 2015), a factor identified previously in studies of the landscape response to the 1980 eruption of Mt St Helens (Collins and Dunne, 1986; Leavesley et al., 1989). Collectively these processes would result in increasing lahar initiation thresholds with time (Van Westen and Daag, 2005).

Probabilistic analysis shows that throughout the two-year dataset utilised in this study, increased 1hrPRI results in increased lahar occurrence probability. Additionally, an increase in the absolute numbers of lahars and a reduction in rain-triggered lahar initiation thresholds are identified in the wet seasons. Seasonality in the nature and frequency of rainfall-generating weather systems controls this pattern. Large lahars are often associated with the passage of synoptic weather systems, which typically produce long-duration catchment-wide rainfall. This is demonstrated by the triggering of large lahars by several named storms during the study dataset including Hurricane Earl in August 2010, Tropical Storm Otto in October 2010 and Tropical Storm Maria in September 2011. Increased rainfall in the wet season also influences antecedent conditions within the catchment, resulting in reduced infiltration rates due to deposit saturation (Barclay et al., 2007). Increased antecedent rainfall can also produce runoff-enhancing surface seals (Segerstrom, 1950; Fohrer et al., 1999) and result in increased bulking efficiency during lahar transit due to high water contents in channel floor deposits (Iverson et al., 2011). These effects increase the overall probability of lahars in the wet season under given rainfall conditions due to flash-flood type responses to rainfall. The reduced frequency of large lahars in the dry season is attributed to the occurrence of fewer sustained catchment-wide synoptic weather systems as well as antecedent effects (low antecedent rainfall inhibits bulking efficiency in the dry season (Fagents and Baloga, 2006; Doyle et al., 2011; Iverson et al., 2011)). The development of lahar magnitude assessment methods, from the subjective classification used in this study, towards quantitative initial flow volume estimates has the potential to enhance probabilistic lahar forecasting by creating probabilistic hazard footprints (Mead et al., 2016). However, such quantitative assessment methods are highly data intensive relative to those developed in this study, requiring pre- and post-

eruption digital elevation models, location specific rainfall intensity-frequency-duration thresholds and physical
deposit characteristics as input data (Mead et al., 2016). These input data requirements prohibit practical
implementation of fully-quantitative magnitude estimates within probabilistic rain-triggered lahar assessment at
all but the most thoroughly monitored volcanoes.
The incorporation of considerations for temporal catchment development and seasonality of prevalent antecedent
conditions into logistic regression-based lahar probability estimation models increases rain-triggered lahar
forecasting performance. The addition of these considerations modulates purely 1hrPRI-based probability
estimates to account for initial deposit moisture content and the degree of catchment recovery during a period of
eruptive quiescence. ROC analysis indicates an excellent ability to differentiate between lahar and non-lahar
outcomes (AUC = 0.83) when only larger rainfall events resulting in ≥8 mm of total rainfall are considered, and
this ability improves even further (AUC = 0.93) when the 8 mm threshold is removed. The readily available model
inputs of 1hrPRI, three-day antecedent rainfall and cumulative rainfall since significant eruptive activity can be
easily assimilated into functional real-time lahar probability estimation models and produces real benefits. Rainfall
gauge networks in volcanic areas are seldom designed with the intention of optimising their usefulness for
detection and characterisation of rain-triggered lahar initiation: the 1hrPRI used in this study is based on the
minimum temporal resolution of the data recorded. Previous studies have shown the utility of 10-minute (Arguden
and Rodolfo, 1990; Tungol and Regalado, 1996; Lavigne et al., 2000; Lavigne and Suwa, 2004; Okano et al.,
2012; Jones et al., 2015), 30-minute (Tungol and Regalado, 1996; Lavigne et al., 2000; Jones et al., 2015) and 60
minute (Lavigne et al., 2000; Lavigne and Suwa, 2004; Jones et al., 2015) rainfall data. Lahar forecasting using
real-time telemetered rainfall data and these techniques has the potential to effectively predict secondary lahars
and increase lahar warning times, even in areas where AFMs, proximal seismometers and trip wires are
unavailable. Used in conjunction with ground-based detectors in instrumented catchments lahar warning times
can be doubled (Jones et al., 2015).
Further research to expand the length of the current two-year study period would develop the understanding of
the catchment recovery-driven temporal trends in lahar occurrence identified within this study. Likewise, the
application of these techniques to additional volcanoes would facilitate both the further examination of the
performance of the lahar forecasting models and the investigation of other important parameters contributing to
the frequency and magnitude of rain-triggered lahar initiation.
**7 Conclusions**
This study demonstrates the development and enhancement of logistic regression-based rain-triggered lahar
probability estimation models for real-time lahar forecasting using the lahar occurrence and rainfall record of the
Belham Valley, Montserrat between April 2010 and April 2012. The incorporation of both antecedent rainfall and
considerations for temporal catchment development into such models alongside the first-order lahar forecasting
parameter of peak rainfall intensity is shown to improve lahar forecasting performance. Rainfall seasonality and
catchment recovery are identified as important factors in the severity of the rain-triggered lahar hazard at Soufrière
Hills Volcano, Montserrat, and by extension similar volcanoes worldwide. Seasonal influences increase both the
absolute number of lahars and the probability of lahar occurrence under pre-defined rainfall conditions during the
wet season due to antecedent effects. Lahar probability is also shown to decline with time under given antecedent
and peak rainfall intensity conditions as a product of catchment evolution. Our results demonstrate the potential
for successful real-time prediction of secondary lahars using readily available input data, even in areas featuring
strongly seasonal climates and periods of eruptive quiescence.
**Competing Interests**
The authors declare that they have no conflict of interest.
**Acknowledgements**
This research was supported by STREVA (NERC/ESRC consortium NE/J02483X/1) and we are thankful to the
Montserrat Volcano Observatory (MVO) for permission to use the lahar database and rain gauge dataset. We
thank Thomas Pierson and Lucia Capra for their constructive reviews which helped improve the paper, and Editor
Thomas Glade.

    **Figure Captions**

**Figure 1: Location map of Montserrat and Soufrière Hills Volcano.**
**Figure 2: Timeline illustrating hourly rainfall data (above) and rain-triggered lahar activity (below) in the Belham**
**Valley, Montserrat between April 2010 and April 2012 (with minor gaps (stippled ornament) due to equipment failure).**
**S, M, and L on the vertical axis represent Small, Medium and Large lahars respectively, see text for details.**
**Figure 3: Timelines displaying examples of lahar triggering rainfall in the Belham Valley, Monserrat between April**
**2010 and April 2012. Alongside the timing of lahar observation and/or detection, the cumulative recorded rainfall (mm)**
**and One Hour Peak Rainfall Intensity (1hrPRI – mm hr$^{-1}$) of the rainfall events are displayed.**
**Figure 4: Illustration of the seasonal fluctuations in lahar occurrence displayed using 6-month data windows with 1-**
**month staggered start dates. Vertical bars indicate the number of lahar events, categorised by magnitude, in each 6-**
**month period. Background contours display the number of rainfall events exceeding specified One Hour Peak Rainfall**
**Intensity (1hrPRI) thresholds, in each 6-month period.**
**Figure 5: Lahar probability, classified by magnitude, as categorised One Hour Peak Rainfall Intensity (1hrPRI)**
**increases. (a) April 2010-April 2012 (b) April 2010-April 2011 (c) April 2011-April 2012.**
**Figure 6: Seasonal and temporal effects on lahar probability. Contour graph of empirically-derived lahar probability**
**relative to the exceedance of One Hour Peak Rainfall Intensity (1hrPRI) thresholds in 6-month moving data windows**
**with 1-month staggered start dates. White numbers and dashed lines show temporal trends. Following the empirically-**
**derived 4 mm hr$^{-1}$ PRI contour, there is a 20% probability of a lahar if this threshold is exceeded at ① (6-month start**
**date of 13/10/2010). This probability increases to 38% at ② (13/04/2011); and declines to 18% at ③ (13/10/2011).**
**Alternatively, reading horizontally across the graph for a lahar probability of 38% the associated PRI threshold**
**increases from 4 mm hr$^{-1}$ at ② (13/04/2011) to approximately 15 mm hr$^{-1}$ at ④ (13/10/2011).**
**Figure 7: Assessment of binary logistic regression-based lahar probability estimation models in the Belham Valley,**
**Montserrat. (a)  Illustration of four binary logistic regression-based lahar probability estimation models created from**
***Year 1*, *Year 2*, *Wet Season* and *Dry Season* data. (b) ROC curves assessing the lahar forecasting performance of an**
**exclusively One Hour Peak Rainfall Intensity (1hrPRI)-centric logistic regression-based lahar probability estimation**
**model and a multi-variable (1hrPRI, antecedent rainfall and long-term cumulative rainfall) model.**

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

Fig.1

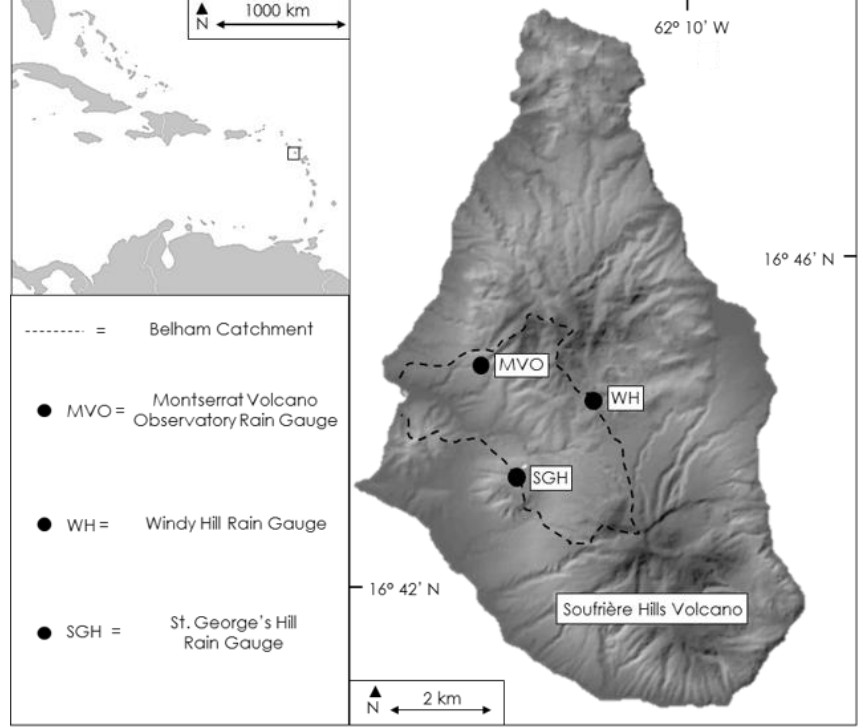


Fig.2

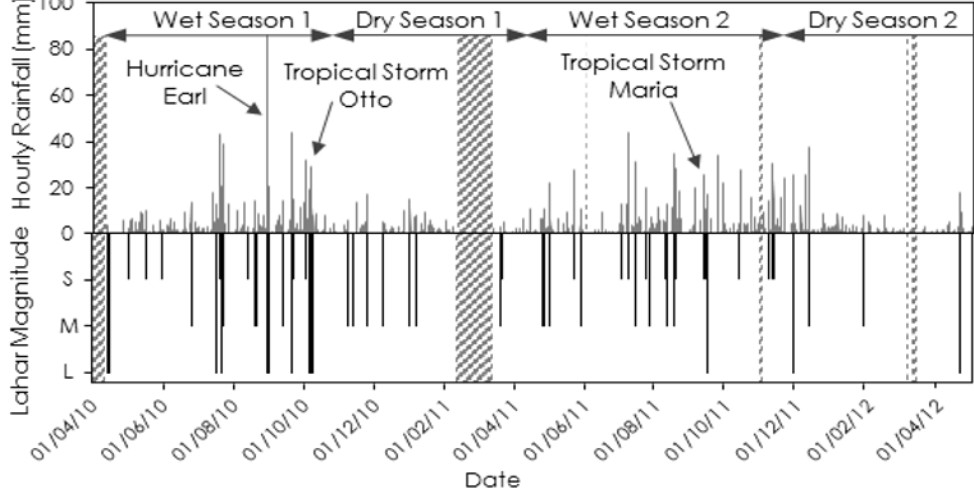

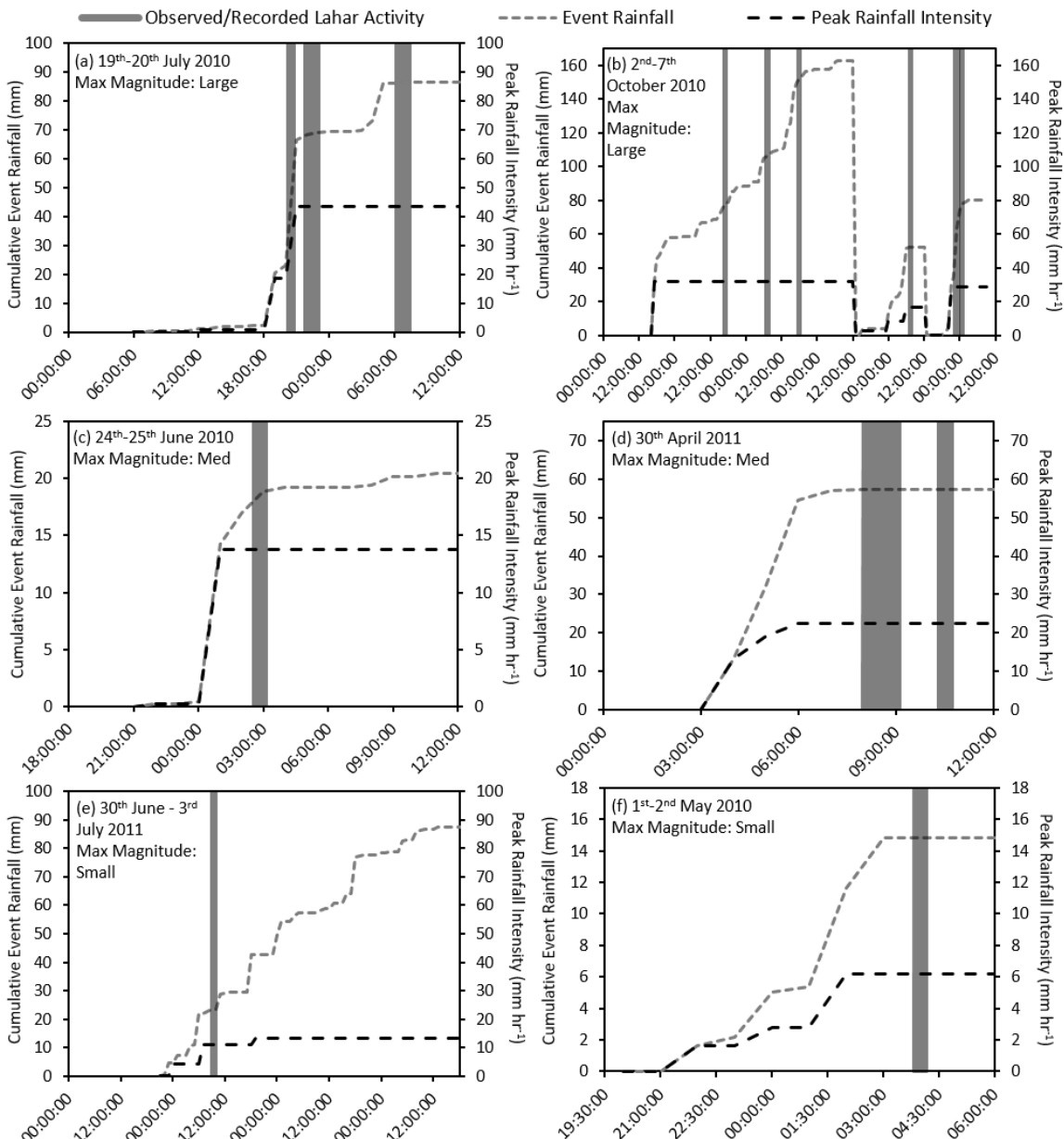

Fig. 4

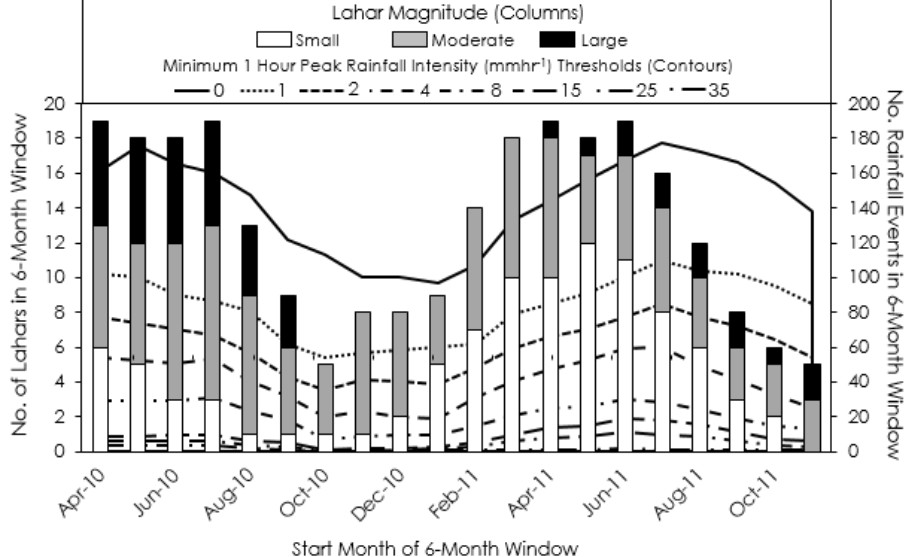


Fig. 5

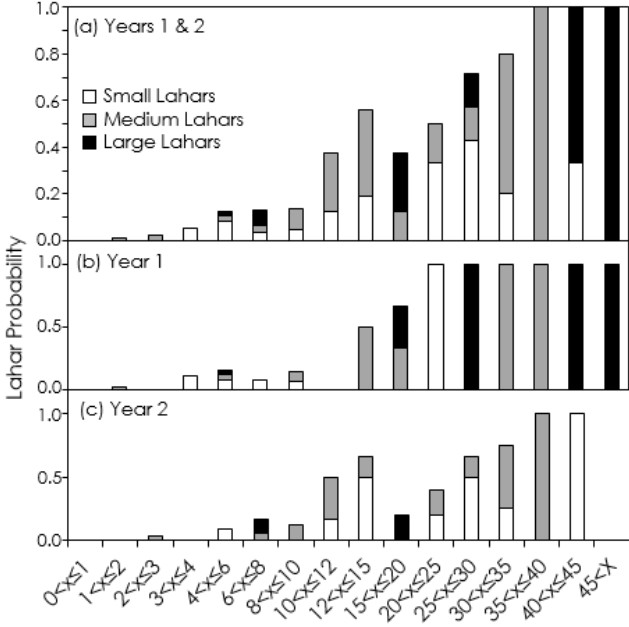


Fig. 6

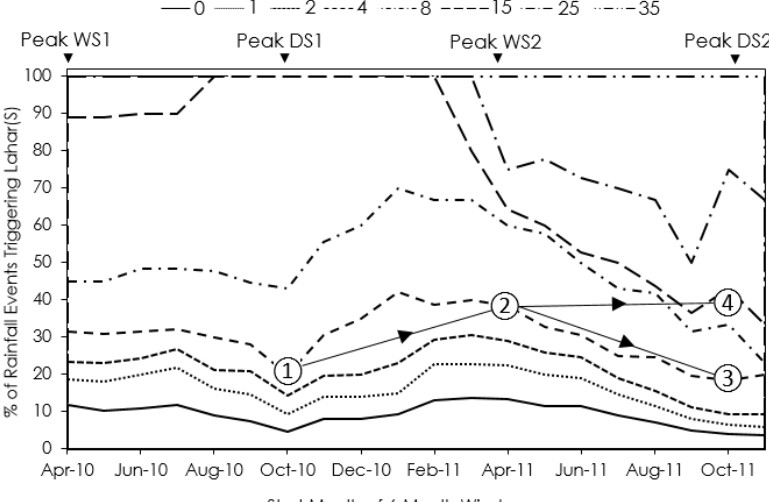


Fig. 7

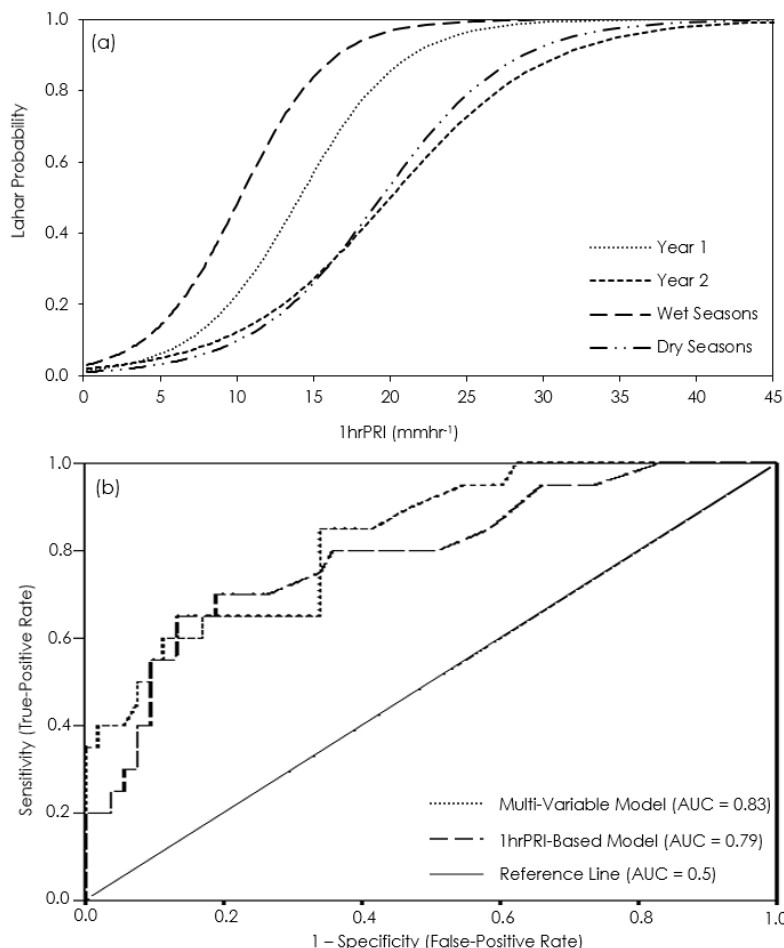


**Appendix I**

Receiver Operating Characteristic (ROC) analysis is a statistical technique that is used to illustrate the diagnostic
ability of a binary classifier system (i.e. a system that subdivides the elements of a given dataset into two groups,
for example the presence or absence of a disease, a pass or a fail in a test etc.). The method was first developed
by electrical and radar engineers during World War II, and has since been used in psychology, medicine,
meteorology, and forecasting of natural hazards.
A graphical plot, or Receiver Operating Characteristics curve (ROC curve) is often used to illustrate the effect of
varying the value of the classifying parameter (for example the number of cancer cells per microlitre of blood or
the pass mark in the previous example). The ROC curve is generated by plotting the true positive rate (TPR)
against the false positive rate (FPR) as the value of the classifying, or threshold parameter, is changed. There are
four possible outcomes from a binary classifier (Table A1): (i) correct prediction of an event that really did occur
= true positive; (ii) incorrect prediction of an event that did not occur = false positive; (iii) predicting no event
when an event does happen = false negative; and (iv) correct prediction that no event occurs and no event really
does occur = true negative.
Imagine a situation where there are 200 patients undergoing a medical test, where alpha is some diagnostic
threshold for having a medical condition. At a given value of alpha, the contingency table could resemble Table
A2.
Here, the TPR is the number of true positives divided by the total number of predicted positives (both true and
false), or $70/(70+30) = 0.70$
The FPR is the number of false positives divided by the total number of predicted negatives (both true and false),
or $28/(28+72) = 0.28$
Thus, for this value of alpha, the corresponding point would plot at $(0.63, 0.28)$ on Figure A1 (the white square).
By systematically varying the value of the threshold parameter alpha, a whole series of 2x2 contingency tables
would be generated, producing an array of points in ROC space and hence a curve (the dashed line).
A 100% rate of prediction (all true positives) would plot at $(0, 1)$ on Figure A1 (the grey circle), whereas a 50%
accurate rate of prediction (i.e. guessing the outcome of a coin toss) would plot at $(0.5, 0.5)$. Random guesses thus
plot along a diagonal line: points above the line represent predictions better than random, points below the line
predictions worse than random.

**Appendix I: Table Captions**

**Table A1: 2x2 contingency table showing the possible outcomes of a binary classifier system.**

**Table A2: 2x2 contingency table for 200 patients undergoing a medical test for the presence or absence of a condition.**

**Appendix I: Figure Captions**

**Fig. A1: ROC space and plots of the prediction examples discussed in the text.**

Table A1

| Total population | Event happens | Event does not happen |
|---|---|---|
| Predict it happens | True positive | False positive |
| Predict it does not happen | False negative | True negative |

Table A2

|                          | Has condition | Has no condition |
|--------------------------|---------------|------------------|
| Predict has condition    | 70            | 30               |
| Predict has no condition | 28            | 72               |

|                          | Has condition | Has no condition |
|--------------------------|---------------|------------------|

Fig. A1

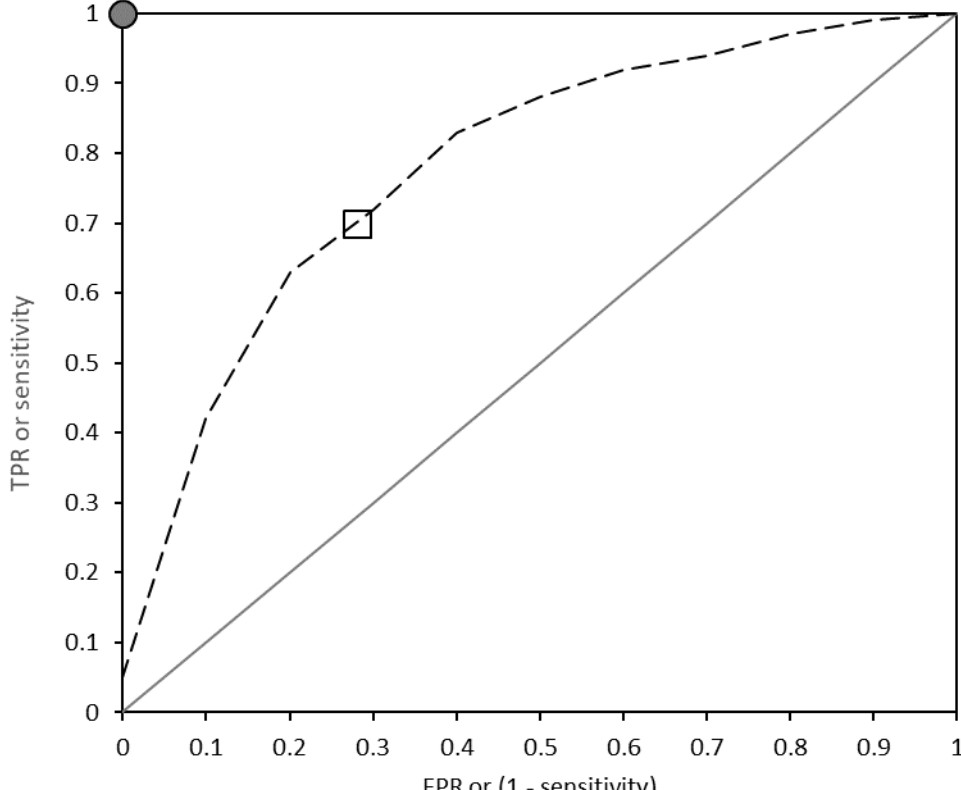
