# Peer review of "Real-time prediction of rain-triggered lahars: incorporating seasonality and catchment recovery"

_Natural Hazards and Earth System Sciences, 2017_

## Referee Comment (RC1) · L. Capra (Referee) · 16 Jun 2017

The paper represents an original contribution aimed to defined lahar occurrence, that represents a very useful tool to be implemented in volcanoes where lahar monitoring systems are not available, or to anticipate the occurrence of an event respect to an early warning system. The model is based on two years records of lahars and their associated rainfalls of the Belham River Valley at Soufrière Hills Volcano, Montserrat. The 1-hour rainfall intensity is used to correlate lahar occurrence in dry and wet season, and lahar probability is defined considering also the 3-day antecedent rainfalls and the catchment evolution. The paper is well organized and nicely illustrated. I have identified some points that need to be better discussed:

[Figure]

A more detailed description of how lahars were grouped in these three different categories is needed (small, medium and large) at least indicating which the main differences are: i.e. duration, magnitude (i.e. maximum amplitude from the seismic record?); runout, flow-depth? Can author also provide a simple description of these lahars, if they are debris flow or hyperconcentraed flow? In addition will be useful to have a table with rainfall characteristics (total accumulated rain, peak intensity) for some selected lahar events, some examples for each lahar category (small, medium, large) in dry and wet season.

Why 1-hour rainfall intensity is here considered? Is a limitation due to the record? I don't know the weather conditions at Monserrat, but in other volcanoes (i.e. Merapi and Colima for example) especially for orographic rains (in the "dry" season), rainfall intensity is calculated over a 5 o 10 min. window, which is much more representative of these type of rains, of short duration (< 1 hours) and high intensity. Do shorter rainfalls (< 1 hrs) have triggered lahars at Montserrat? Is 1-hour peak intensity representative of different rainfall behaviors at Montserrat? Would you expect any difference in your model with a 10-min. peak rainfall intensity?

Line 116. How the 1-hourPRIs threshold is defined?

Line 124-129. From figure 2 at least two large lahars occurred in the dry seson2, with accumulated rainfall less than 20 mm for at least one of them. There are any evidences of hydrophobicity? Which type of vegetation grows at Soufriere Hills volcano? In addition, small lahars are more common in the wet season. For example during dry seasons 1 and 2 only medium (and 2 large) lahars were recorded and small events are only observed in the wet season. Please add some consideration about this behavior in the discussion section, at line 215-218.

Line 140-141. "This indicates that more intense rainfall is required to trigger lahars in the dry season 141 than in the wet season." Can author please discuss this behavior? Is this correlated with a higher permeability of the substratum in the dry season? How

much rains accumulate during these high intensity events in the dry season?

Line 165: 3-day antecedent rainfall values is a common time interval also used in previous works, such as at Colima volcano, please add some references.

Line 166. Can authors be more specific about the definition of the term "total cumulative rainfall since significant eruptive activity"? In their model will be the total rain since Phase 5? And, how this term reflect the catchment evolution?

Line 215-218. This point needs a better discussion in light of Figure 2 (see previous comment at line 124-129).

Line 225-227. This is questionable based on data here presented; see previous comment about figure 2.

---

## Referee Comment (RC2) · T. Pierson (Referee) · 4 Jul 2017

Attempts to use rainfall intensity/duration thresholds to effectively predict debris-flow occurrence in non-volcanic terrains and lahar occurrence in volcanic landscapes have been ongoing for decades. This paper, utilizing a rich data set from Montserrat and innovative statistical treatments of the data, makes an important contribution to the discussion. The paper is clearly and concisely written and the figures are quite good. Overall, I would like to see a bit more clarification of the methods used, more explanation (in plain English) of what the statistical treatments are attempting to show, and a broader discussion of the significance of the results in the context of other research.

What makes this paper an important contribution is the authors' consideration of (1)

catchment stability (measured as total cumulative rainfall since the last significant eruptive activity);and (2) the "false positives problem", i.e., when the occurrence of rainfall intensities above a threshold can, in some cases, trigger lahars but which in other cases do not. While the conclusions reached on both of these topics are a valuable contribution, more discussion of the significance of these findings in the context of previous studies would be extremely helpful.

There are several places in the paper where more attention is needed to clarify the research itself and its significance: 1) It would be helpful if there were a Methods section that summarized all of the approaches and assumptions used in the study. Explanations of these are currently scattered throughout the paper. 2) The sentence in lines 52–56 is overly complex and confusing. In fact, a word seems to be missing. 3) In line 64 it would be good to say a bit more about what is meant by "temporal catchment development." 4) In lines 81 and 84 there is inconsistent capitalization of "Vulcanian." 5) At the beginning of section 4, please explain why data sets from different rain gauges are used for different time intervals. Different catch efficiencies can bias results between gauges, and local convective rainstorms can deliver different RF amounts to different gauges. 6) More explanation is needed for how the peak rainfall intensity (PRI) of 1 hour was chosen for the analyses, and some discussion of PRIs used by other researchers is warranted. 7) What are the time lags between the PRIs and lahar initiations? 8) Decline in lahar frequency and magnitude following catchment disturbance is a commonly reported phenomenon. Discussion is needed on how the results of this study specifically compare to the results of other studies. 9) Sentence in lines 187–189 is unclear. Is there a word missing? 10) In lines 193–194, the AUC produced by Eq. 2 is given for the analysis of all RF events. What is it for Eq. 3? 11) Discussion is needed for why the antecedent moisture index of 3-day previous rainfall was chosen. What indices have been used by other researchers? 12) In lines 225–226, it would seem that the longer durations of the synoptic rainstorms are critical for providing the antecedent moisture during the wet season. It would be good to emphasize that here for the main reason that lahars are harder to trigger in the dry season. 13) In line 227,

a reference for inefficient bulking in dry channels is in order. 14) Toward the end of the discussion section, a better explanation of the meaning and significance of the ROC analysis is needed. From what you have written, I assume (not being familiar with this analysis) that (1) AUC = 0.5 means the number of true positives equals the number of false positives, and that (2) AUC = 1.0 means the number of true positives is 100%. Is this the case? 15) How far above the PRI thresholds are the false-positive rainfall intensities? For example, if you set a PRI threshold of 25 mm/hr, how large a PRI can occur that does not trigger a lahar? 16) Figure 2 caption: Please explain the vertical dashed lines.

---

## Author Comment (AC1) · 29 Aug 2017

We would like to thank the reviewer for the comments and constructive suggestions relating to the underlying review of manuscript number nhess-2017-166. Please find below the reviewer's comment and authors' replies to these comments:

L. Capra (Referee) lcapra@geociencias.unam.mx

Comment: The paper represents an original contribution aimed to defined lahar occurrence, that represents a very useful tool to be implemented in volcanoes where lahar monitoring systems are not available, or to anticipate the occurrence of an event respect to an early warning system. The model is based on two years records of lahars

and their associated rainfalls of the Belham River Valley at Soufrière Hills Volcano, Montserrat. The 1-hour rainfall intensity is used to correlate lahar occurrence in dry and wet season, and lahar probability is defined considering also the 3-day antecedent rainfalls and the catchment evolution. The paper is well organized and nicely illustrated. I have identified some points that need to be better discussed:

Comment: A more detailed description of how lahars were grouped in these three different categories is needed (small, medium and large) at least indicating which the main differences are: i.e. duration, magnitude (i.e. maximum amplitude from the seismic record?); runout, flow-depth?

Reply: Increased information can be included in the manuscript regarding the magnitude categories assigned to the lahars. These categories were assessed using visual inspection of the degree of channel inundation and flow depth (where possible); in addition to the assessment of the duration and amplitude of seismic signals. Lahar signals show continuous readings in the 2-5 hz and peak at approximately 30 hz. The highest recorded amplitudes are associated with discharge and solid load in the lahar (based on observations). Lahar signals were cross referenced to visual observations and carefully excluded from signals associated with primary activity and other seismic noise (such as construction vehicles).

Comment: Can author also provide a simple description of these lahars, if they are debris flow or hyperconcentrated flow?

Reply: Detailed observations of lahars in the Belham River Valley have indicated that they are Newtonian and fully turbulent (Barclay et al., 2007; Susnik, 2009; Alexander et al., 2010; Froude et al., 2017) This interpretation is based on sampling of several small and large events and two detailed studies of flow deposits (2006-2009 and 2012-2015). Further details may be provided, however detailed observations of a flow and associated previous studies are fully referenced in Froude et al. (2017).

Comment: In addition will be useful to have a table with rainfall characteristics (total

accumulated rain, peak intensity) for some selected lahar events, some examples for each lahar category (small, medium, large) in dry and wet season.

Reply: The authors agree and feel that a multi-part figure illustrating the timeline of several rainfall events and the associated lahar activity (size, timing and duration) and rainfall characteristics (timing, cumulative rainfall and peak intensity) could be added to the manuscript and would be of significant benefit to the research.

Comment: Why 1-hour rainfall intensity is here considered? Is a limitation due to the record? I don't know the weather conditions at Monserrat, but in other volcanoes (i.e. Merapi and Colima for example) especially for orographic rains (in the "dry" season), rainfall intensity is calculated over a 5 o 10 min. window, which is much more representative of these type of rains, of short duration (< 1 hours) and high intensity. Do shorter rainfalls (< 1 hrs) have triggered lahars at Montserrat? Is 1-hour peak intensity representative of different rainfall behaviors at Montserrat? Would you expect any difference in your model with a 10-min. peak rainfall intensity?

Reply: The reviewer is correct in identifying that 1-hour rainfall intensity was utilised in this study due to a limitation of the record (it was the maximum temporal resolution available). As noted by the reviewer, at other locations including Colima, Merapi and Tungurahua, 10-minute rainfall has been utilised and this has benefits in terms of assessing lahar triggering rainfall from short-duration high-intensity rainfall events which frequently occur in the tropics (e.g. Lavigne & Suwa, 2004; Capra et al. 2010; Jones et al. 2015). Short duration rainfall has resulted in lahars in the Belham Valley within the studied dataset and increased temporal rainfall data resolution would certainly be advantageous if available. However, the 1-hour approach has been demonstrated to be an effective basis for the methods developed in this study (Lavigne et al. 2000; Lavigne & Suwa, 2004; Jones et al. 2015). If incorporated alongside the current 1-hour peak rainfall intensity, 10-minute rainfall intensity could potentially be expected to further increase model performance by more appropriately capturing lahars triggered by short duration, high-intensity events. A discussion point relating to this concept could

be added to the manuscript.

Comment: Line 116. How the 1-hourPRIs threshold is defined?

Reply: In this study 1-hour peak rainfall intensity is defined as the maximum rainfall recorded in one hour during a single rainfall event. A single rainfall event is defined as a period of recorded rainfall in between two dry spells of six hours or longer. The 1-hour PRI thresholds referred to in the manuscript separate the dataset into those rainfall events which exceeded a given peak intensity threshold and those which did not, and examines the rate of lahar occurrence in each case. More detail regarding these definitions can be incorporated into the manuscript for clarity.

Comment: Line 124-129. From figure 2 at least two large lahars occurred in the dry seson2, with accumulated rainfall less than 20 mm for at least one of them. There are any evidences of hydrophobicity? Which type of vegetation grows at Soufriere Hills volcano?

Reply: Prior to the onset of eruptive activity 62% of the Belham Catchment was densely vegetated with Dry Forest (29%), Mesic Forest (48%) and Wet Forest (13%), with dry forest subsequently identified as the dominant species found on re-vegetating pyroclastic deposits (Froude 2015). Previous studies in the Belham Valley have not identified evidence of hydrophobicity, such as previously identified at Colima by Capra et al. (2010). In the Belham Valley increased vegetation damage has been identified as increasing lahar occurrence (Barclay et al, 2007; Alexander et al, 2010) and increased lahar activity late in the wet season attributed to increased deposit saturation and decreased infiltration rates (Barclay et al, 2007). Figure 2 displays hourly rainfall and whilst it is correct that neither of the two large lahars in dry season two were triggered by rainfall events featuring 1-hour PRI values of >20 mmhr-1, they were associated with rainfall events with significant total rainfall values of 39 mm (29/11/2011) and 22 mm (19/04/2012).

Comment: In addition, small lahars are more common in the wet season. For example

during dry seasons 1 and 2 only medium (and 2 large) lahars were recorded and small events are only observed in the wet season. Please add some consideration about this behaviour in the discussion section, at line 215-218.

Reply: Small events are indeed more common in the wet season, a factor attributed to "flash flood" responses to rainfall during periods of increased antecedent rainfall. Small magnitude pulses of lahar activity did occur due to rainfall during dry seasons 1 and 2, however these often occurred during rainfall events which also triggered larger magnitude pulses and as such the small pulses are superseded in Figure 2.

Comment: Line 140-141. "This indicates that more intense rainfall is required to trigger lahars in the dry season than in the wet season." Can author please discuss this behaviour? Is this correlated with a higher permeability of the substratum in the dry season? How much rains accumulate during these high intensity events in the dry season?

Reply: The dataset indicated that lahars were statistically more likely to be triggered for a given peak rainfall intensity in the wet season compared to the dry season. This is thought to be a product of increased infiltration rates in the dry season associated with generally lower levels of antecedent rainfall. In terms of individual dry-season rainfall events that did not trigger lahars (of sufficient magnitude to be detected on the seismic records); 64 mm of rainfall was recorded on 4th/5th January 2011 and 73 mm on 4th/5th December 2011 without any recorded lahars. Recorded 3-Day antecedent rainfall was less than 3.1 mm at the onset of both rainfall events.

Comment: Line 165: 3-day antecedent rainfall values is a common time interval also used in previous works, such as at Colima volcano, please add some references.

Reply: Absolutely, additional references including Capra et al. (2010) to the prior use of 3-day antecedent rainfall will be added. Information and references will also be included regarding the previous use of other timescales (including 24-hour and 7-day antecedent rainfall) and how 3-day rainfall was chosen as the optimal timescale within

this study.

Comment: Line 166. Can authors be more specific about the definition of the term "total cumulative rainfall since significant eruptive activity"? In their model will be the total rain since Phase 5? And, how this term reflect the catchment evolution?

Reply: The reviewer is correct, the term "total cumulative rainfall since significant activity" reflects the total rainfall since the end of Phase 5. This parameter is used as a proxy for catchment evolution within the model under the assumption that in the absence of further eruptive activity hydrogeomorphic drainage basin recovery will occur following the catchment disturbance associated with phase 5 (Pierson & Major, 2014).

Comment: Line 215-218. This point needs a better discussion in light of Figure 2 (see previous comment at line 124-129).

Reply: As the reviewer identifies in their comment relating to line 124-129, large lahars are not exclusively triggered in the wet season and there are examples of large lahars in the dry season. However, the primary objective of the point in lines 215-218 is to emphasise that large lahars are frequently associated with the passage of large synoptic weather systems which produce large volumes of total rainfall. The increased frequency of rainfall events in the wet season (including such synoptic systems) results in an increase in the average antecedent rainfall, which is identified as contributing to the observed reduction in 1hr PRI based lahar initiation thresholds during the wet season.

Comment: Line 225-227. This is questionable based on data here presented; see previous comment about figure 2.

Reply: As identified by the reviewer, the term "absence of large lahars in the dry season" should be replaced with "the reduction in the frequency of large lahars in the dry season" as there are a couple of examples of such flows within the studied dataset. However, this reduction is still attributed to a combination of the occurrence of fewer

sustained catchment-wide synoptic weather systems and a reduction in average antecedent rainfall and thus saturation level of pyroclastic deposits.

References: Alexander, J., Barclay, J., Susnik, J., Loughlin, S. C., Herd, R. A., Darnell, A., and Crosweller, S.: Sediment-charged flash floods on Montserrat: The influence of synchronous tephra fall and varying extent of vegetation damage, Journal of Volcanology and Geothermal Research, 194, 127-138, 10.1016/j.jvolgeores.2010.05.002, 2010.

Barclay, J., Alexander, J., and Susnik, J.: Rainfall-induced lahars in the Belham Valley, Montserrat, West Indies, Journal of the Geological Society, 164, 815-827, 10.1144/0016-76492006-078, 2007.

Capra, L., Borselli, L., Varley, N., Gavilanes-Ruiz, J. C., Norini, G., Sarocchi, D., Caballero, L., and Cortes, A.: Rainfall-triggered lahars at Volcán de Colima, Mexico: Surface hydro-repellency as initiation process, Journal of Volcanology and Geothermal Research, 189, 105-117, 10.1016/j.jvolgeores.2009.10.014, 2010.

Froude, M. J.: Lahar Dynamics in the Belham River Valley, Montserrat: Application of Remote Camera-Based Monitoring for Improved Sedimentological Interpretation of Post-Event Deposits, PhD Thesis, School of Environmental Science, University of East Anglia, 2015.

Froude, M.J., Alexander, A., Barclay, J., Cole, P. (2017) Interpreting flash flood paleoflow parameters from antidunes and gravel lenses: An example from Montserrat, West Indies, Sedimentology, DOI:10.1111/sed.12375

Jones, R., Manville, V., and Andrade, D.: Probabilistic analysis of rain-triggered lahar initiation at Tungurahua volcano, Bulletin of Volcanology, 77, 10.1007/s00445-015-0946-7, 2015.

Lavigne, F., Thouret, J. C., Voight, B., Young, K., LaHusen, R., Marso, J., Suwa, H., Sumaryono, A., Sayudi, D. S., and Dejean, M.: Instrumental lahar monitoring at Merapi

Volcano, Central Java, Indonesia, Journal of Volcanology and Geothermal Research, 100, 457-478, 10.1016/S0377-0273(00)00151-7, 2000.

Lavigne, F., and Suwa, H.: Contrasts between debris flows, hyperconcentrated flows and stream flows at a channel of Mount Semeru, East Java, Indonesia, Geomorphology, 61, 41-58, 10.1016/j.geomorph.2003.11.005, 2004.

Pierson, T. C., and Major, J. J.: Hydrogeomorphic effects of explosive volcanic eruptions on drainage basins, Annual Review of Earth and Planetary Sciences, 42, 469-507, 10.1146/annurev-earth-060313-054913, 2014.

---

## Author Comment (AC2) · 29 Aug 2017

Many thanks to the reviewer for the comments and constructive suggestions relating to the underlying review of manuscript number nhess-2017-166. Please find below the reviewer's comments and the authors' replies to these comments:

T. Pierson (Referee) tpierson@usgs.gov

Comment: Attempts to use rainfall intensity/duration thresholds to effectively predict debris-flow occurrence in non-volcanic terrains and lahar occurrence in volcanic landscapes have been ongoing for decades. This paper, utilizing a rich data set from Montserrat and innovative statistical treatments of the data, makes an important contribution to the discussion. The paper is clearly and concisely written and the figures

are quite good. Overall, I would like to see a bit more clarification of the methods used, more explanation (in plain English) of what the statistical treatments are attempting to show, and a broader discussion of the significance of the results in the context of other research. What makes this paper an important contribution is the authors' consideration of (1) catchment stability (measured as total cumulative rainfall since the last significant eruptive activity); and (2) the "false positives problem", i.e., when the occurrence of rainfall intensities above a threshold can, in some cases, trigger lahars but which in other cases do not. While the conclusions reached on both of these topics are a valuable contribution, more discussion of the significance of these findings in the context of previous studies would be extremely helpful.

There are several places in the paper where more attention is needed to clarify the research itself and its significance:

1) It would be helpful if there were a Methods section that summarized all of the approaches and assumptions used in the study. Explanations of these are currently scattered throughout the paper.

Reply: The authors agree that a restructure of the manuscript to include a consolidated methods section would be beneficial to the manuscript.

2) The sentence in lines 52–56 is overly complex and confusing. In fact, a word seems to be missing.

Reply: Amendments to this sentence are required and would help to clarify this section. E.g. "Despite this geographic coincidence and the importance of climatic rainfall regimes on storm intensities, durations and antecedent conditions (all significant factors in lahar initiation: Pierson and Major (2014)), the impact of seasonal rainfall on rain-triggered lahar initiation has not previously been explicitly considered within the development of rain-triggered lahar hazard assessment tools."

3) In line 64 it would be good to say a bit more about what is meant by "temporal

catchment development."

Reply: Absolutely, this is a key theme later in the manuscript and it would be beneficial to further develop the introduction to this topic at this point in the manuscript. Studies including but not limited to Major et al. (2000), Major & Yamakoshi (2005), Gran & Montgomery (2005) and Pierson & Major (2014) extensively cover this topic and could be used to provide key references when developing this concept within the manuscript.

4) In lines 81 and 84 there is inconsistent capitalization of "Vulcanian."

Reply: This inconsistency will be rectified.

5) At the beginning of section 4, please explain why data sets from different rain gauges are used for different time intervals. Different catch efficiencies can bias results between gauges, and local convective rainstorms can deliver different RF amounts to different gauges.

Reply: The different rain gauges were used for different time periods out of necessity, and it would indeed be advantageous to have both enhanced continuity of rain gauge location and increased spatial distribution of rainfall gauges across the catchment. As highlighted by the reviewer, the spatial variability in recorded rainfall from local convective rainstorms is certainly a consideration in the Belham Valley. However, the methods presented in this manuscript using the different rain gauges are shown to effectively forecast lahars, and this effectiveness could potentially be further enhanced at locations where networks of permanent gauges are present. Equipment failure is a common issue in monitoring volcanic environments and it of potential benefit that the method here is robust against this.

6) More explanation is needed for how the peak rainfall intensity (PRI) of 1 hour was chosen for the analyses, and some discussion of PRIs used by other researchers is warranted.

Reply: One hour peak rainfall intensity was the highest temporal resolution available

and as such was the selected resolution. Other studies have shown one-hour peak rainfall intensity to be an effective parameter in lahar initiation threshold assessment (e.g. Jones et al. 2015), although if higher temporal resolutions were available these would have the potential to enhance the performance of lahar forecasting tools, particularly with respect to more accurately capturing the intensities of local convective rainfall events. Previous studies have shown 10-minute rainfall (Arguden & Rodolfo, 1990; Tungol & Regalado, 1996; Lavigne et al. 2000; Lavigne & Suwa, 2004; Okano et al. 2012, Jones et al. 2015), 30-minute rainfall (Lavigne et al. 2000; Tungol & Regalado, 1996; Jones et al. 2015) and 1 hour rainfall (Lavigne et al. 2000; Lavigne & Suwa, 2004; Jones et al. 2015) to be useful parameters in the assessment of lahar hazard.

7) What are the time lags between the PRIs and lahar initiations?

Reply: The authors agree that highlighting the lag time between recorded rainfall and lahar detection is important in portraying the potential benefits of the methods discussed in this manuscript. Examples of lag times will be displayed in a new figure displaying the timelines of individual lahar events and recorded rainfall data.

8) Decline in lahar frequency and magnitude following catchment disturbance is a commonly reported phenomenon. Discussion is needed on how the results of this study specifically compare to the results of other studies.

Reply: A decline in lahar frequency following catchment disturbance is indeed a commonly reported phenomenon, although direct comparison of the results of this study to previous research is difficult due to the contrasting methods used. However, general comparisons of the conclusions of studies including Van Westen & Daag (2005), which identify increasing lahar initiation thresholds with time, would be beneficial to the manuscript.

9) Sentence in lines 187–189 is unclear. Is there a word missing?

Reply: The authors agree that this sentence could be amended to improve its clarity.

E.g. "ROC analysis plots the true positive rate against the false positive rate as a threshold (estimated lahar probability in this instance) is varied in order to assess how effectively the parameter discriminates between lahar and non-lahar producing rainfall events."

10) In lines 193–194, the AUC produced by Eq. 2 is given for the analysis of all RF events. What is it for Eq. 3?

Reply: The AUC produced by Eq. 3 is 0.89 when all rainfall events are analysed, indicating that the AUC increases by a similar magnitude to that of Eq. 2 when all rainfall events (regardless of magnitude) are considered. This detail can be added to the manuscript.

11) Discussion is needed for why the antecedent moisture index of 3-day previous rainfall was chosen. What indices have been used by other researchers?

Reply: A key point also raised by another reviewer, the discussion of the use of antecedent rainfall by other researchers will be expanded and specific mention will be given as to why 3-day rainfall was selected alongside other timescales for testing as an antecedent moisture index. When tested within this study, 3-day antecedent was the optimal timescale, as also utilised by Capra et al. (2010) at Colima, where the lower rainfall and higher evaporation rates made this shorter timescale more relevant than the 7-day timescale used in previous studies in Indonesia (Lavigne et al. 2000; Lavigne & Suwa 2004). As well as being heavily influenced by local climate (Capra et al. 2010), the optimal antecedent rainfall timescale is also influenced by the grain size of pyroclastic material in lahar source regions (Rodolfo & Arguden, 1991). 24-hour (Okano et al. 2012; Jones et al. 2015), 3-day (Capra et al. 2010; Jones et al. 2015) and 7-day (Lavigne et al. 2000; Lavigne & Suwa, 2004) antecedent rainfall have been used in previous research as a lahar initiation threshold assessment parameter.

12) In lines 225–226, it would seem that the longer durations of the synoptic rainstorms are critical for providing the antecedent moisture during the wet season. It would be

good to emphasize that here for the main reason that lahars are harder to trigger in the dry season.

Reply: An excellent point and a topic that needs to be further emphasised in the manuscript. The total volume of rainfall applied during the wet season during synoptic events is key to decreasing lahar initiation thresholds.

13) In line 227, a reference for inefficient bulking in dry channels is in order.

Reply: The authors agree, references to this process will be added to the manuscript, including Fagents & Baloga (2006), Doyle et al. (2011) and others.

14) Toward the end of the discussion section, a better explanation of the meaning and significance of the ROC analysis is needed. From what you have written, I assume (not being familiar with this analysis) that (1) AUC = 0.5 means the number of true positives equals the number of false positives, and that (2) AUC = 1.0 means the number of true positives is 100%. Is this the case?

Reply: This understanding of ROC analysis is correct, however further explanation of ROC analysis would be beneficial to the manuscript and could be implemented within the proposed updated methods section.

15) How far above the PRI thresholds are the false-positive rainfall intensities? For example, if you set a PRI threshold of 25 mm/hr, how large a PRI can occur that does not trigger a lahar?

Reply: Taking the reviewer's example, if a strict threshold of 25 mm/hr was selected there would be 18 rainfall events in the study period above this threshold that would be expected to trigger lahars. Of these 18 rainfall events, there would be three false positives, with peak rainfall intensities of 26, 28 and 34 mm/hr respectively. All rainfall events exceeding 34 mm/hr that were analysed in this study triggered lahars. Consideration of this topic could be added to the manuscript as a discussion point.

16) Figure 2 caption: Please explain the vertical dashed lines.

Reply: These dashed lines are periods where equipment failure occurred and resulted in a gap in the record. Further detail will be added to the caption to make this clearer.

References:

[revised manuscript text omitted]

---

## Author Response (AR1)

We would like to thank the reviewer for the comments and constructive suggestions relating to the underlying
review of manuscript number *nhess-2017-166*. Please find below the authors' replies (in blue italics) to each of
these comments:

**L. Capra (Referee)**

lcapra@geociencias.unam.mx

The paper represents an original contribution aimed to defined lahar occurrence, that represents a very useful tool
to be implemented in volcanoes where lahar monitoring systems are not available, or to anticipate the occurrence
of an event respect to an early warning system. The model is based on two years records of lahars and their
associated rainfalls of the Belham River Valley at Soufrière Hills Volcano, Montserrat. The 1-hour rainfall
intensity is used to correlate lahar occurrence in dry and wet season, and lahar probability is defined considering
also the 3-day antecedent rainfalls and the catchment evolution. The paper is well organized and nicely illustrated.

I have identified some points that need to be better discussed:

A more detailed description of how lahars were grouped in these three different categories is needed (small,
medium and large) at least indicating which the main differences are: i.e. duration, magnitude (i.e. maximum
amplitude from the seismic record?); runout, flow-depth?

• *Increased information can be included in the manuscript regarding the magnitude categories assigned*
*to the lahars. These categories were assessed using visual inspection of the degree of channel inundation*
*and flow depth (where possible); in addition to the assessment of the duration and amplitude of seismic*
*signals. Lahar signals show continuous readings in the 2-5 hz and peak at approximately 30 hz. The*
*highest recorded amplitudes are associated with discharge and solid load in the lahar (based on*
*observations). Lahar signals were cross referenced to visual observations and carefully excluded from*
*signals associated with primary activity and other seismic noise (such as construction vehicles).*

Can author also provide a simple description of these lahars, if they are debris flow or hyperconcentrated flow?

• *Detailed observations of lahars in the Belham River Valley have indicated that they are Newtonian and*
*fully turbulent (Barclay et al., 2007; Susnik, 2009; Alexander et al., 2010; Froude et al., 2017) This*
*interpretation is based on sampling of several small and large events and two detailed studies of flow*
*deposits (2006-2009 and 2012-2015). Further details may be provided, however detailed observations*
*of a flow and associated previous studies are fully referenced in Froude et al. (2017).*

In addition will be useful to have a table with rainfall characteristics (total accumulated rain, peak intensity) for
some selected lahar events, some examples for each lahar category (small, medium, large) in dry and wet season.

• *The authors agree and feel that a multi-part figure illustrating the timeline of several rainfall events and*
*the associated lahar activity (size, timing and duration) and rainfall characteristics (timing, cumulative*
*rainfall and peak intensity) could be added to the manuscript and would be of significant benefit to the*
*research.*

Why 1-hour rainfall intensity is here considered? Is a limitation due to the record? I don't know the weather
conditions at Monserrat, but in other volcanoes (i.e. Merapi and Colima for example) especially for orographic
rains (in the "dry" season), rainfall intensity is calculated over a 5 o 10 min. window, which is much more
representative of these type of rains, of short duration (< 1 hours) and high intensity. Do shorter rainfalls (< 1 hrs)
have triggered lahars at Montserrat? Is 1-hour peak intensity representative of different rainfall behaviors at
Montserrat? Would you expect any difference in your model with a 10-min. peak rainfall intensity?

• *The reviewer is correct in identifying that 1-hour rainfall intensity was utilised in this study due to a*
*limitation of the record (it was the maximum temporal resolution available). As noted by the reviewer,*
*at other locations including Colima, Merapi and Tungurahua, 10-minute rainfall has been utilised and*
*this has benefits in terms of assessing lahar triggering rainfall from short-duration high-intensity rainfall*
*events which frequently occur in the tropics (e.g. Lavigne & Suwa, 2004; Capra et al. 2010; Jones et al.*
*2015). Short duration rainfall has resulted in lahars in the Belham Valley within the studied dataset and*
*increased temporal rainfall data resolution would certainly be advantageous if available. However, the*
*1-hour approach has been demonstrated to be an effective basis for the methods developed in this study*
*(Lavigne et al. 2000; Lavigne & Suwa, 2004; Jones et al. 2015). If incorporated alongside the current*
*1-hour peak rainfall intensity, 10-minute rainfall intensity could potentially be expected to further*

*increase model performance by more appropriately capturing lahars triggered by short duration, high-*
*intensity events. A discussion point relating to this concept could be added to the manuscript.*

Line 116. How the 1-hourPRIs threshold is defined?

• *In this study 1-hour peak rainfall intensity is defined as the maximum rainfall recorded in one hour*
*during a single rainfall event. A single rainfall event is defined as a period of recorded rainfall in between*
*two dry spells of six hours or longer. The 1-hour PRI thresholds referred to in the manuscript separate*
*the dataset into those rainfall events which exceeded a given peak intensity threshold and those which*
*did not, and examines the rate of lahar occurrence in each case. More detail regarding these definitions*
*can be incorporated into the manuscript for clarity.*

Line 124-129. From figure 2 at least two large lahars occurred in the dry seson2, with accumulated rainfall less
than 20 mm for at least one of them. There are any evidences of hydrophobicity? Which type of vegetation grows
at Soufriere Hills volcano?

• *Prior to the onset of eruptive activity 62% of the Belham Catchment was densely vegetated with Dry*
*Forest (29%), Mesic Forest (48%) and Wet Forest (13%), with dry forest subsequently identified as the*
*dominant species found on re-vegetating pyroclastic deposits (Froude 2015). Previous studies in the*
*Belham Valley have not identified evidence of hydrophobicity, such as previously identified at Colima*
*by Capra et al. (2010). In the Belham Valley increased vegetation damage has been identified as*
*increasing lahar occurrence (Barclay et al, 2007; Alexander et al, 2010) and increased lahar activity*
*late in the wet season attributed to increased deposit saturation and decreased infiltration rates (Barclay*
*et al, 2007). Figure 2 displays hourly rainfall and whilst it is correct that neither of the two large lahars*
*in dry season two were triggered by rainfall events featuring 1-hour PRI values of >20 mmhr$^{-1}$, they*
*were associated with rainfall events with significant total rainfall values of 39 mm (29/11/2011) and 22*
*mm (19/04/2012).*

In addition, small lahars are more common in the wet season. For example during dry seasons 1 and 2 only medium
(and 2 large) lahars were recorded and small events are only observed in the wet season. Please add some
consideration about this behaviour in the discussion section, at line 215-218.

• *Small events are indeed more common in the wet season, a factor attributed to "flash flood" responses*
*to rainfall during periods of increased antecedent rainfall. Small magnitude pulses of lahar activity did*
*occur due to rainfall during dry seasons 1 and 2, however these often occurred during rainfall events*
*which also triggered larger magnitude pulses and as such the small pulses are superseded in Figure 2.*

Line 140-141. "This indicates that more intense rainfall is required to trigger lahars in the dry season than in the
wet season." Can author please discuss this behaviour? Is this correlated with a higher permeability of the
substratum in the dry season? How much rains accumulate during these high intensity events in the dry season?

• *The dataset indicated that lahars were statistically more likely to be triggered for a given peak rainfall*
*intensity in the wet season compared to the dry season. This is thought to be a product of increased*
*infiltration rates in the dry season associated with generally lower levels of antecedent rainfall. In terms*
*of individual dry-season rainfall events that did not trigger lahars (of sufficient magnitude to be detected*
*on the seismic records); 64 mm of rainfall was recorded on 4$^{th}$/5$^{th}$ January 2011 and 73 mm on 4$^{th}$/5$^{th}$*
*December 2011 without any recorded lahars. Recorded 3-Day antecedent rainfall was less than 3.1 mm*
*at the onset of both rainfall events.*

Line 165: 3-day antecedent rainfall values is a common time interval also used in previous works, such as at
Colima volcano, please add some references.

• *Absolutely, additional references including Capra et al. (2010) to the prior use of 3-day antecedent*
*rainfall will be added. Information and references will also be included regarding the previous use of*
*other timescales (including 24-hour and 7-day antecedent rainfall) and how 3-day rainfall was chosen*
*as the optimal timescale within this study.*

Line 166. Can authors be more specific about the definition of the term "total cumulative rainfall since significant
eruptive activity"? In their model will be the total rain since Phase 5? And, how this term reflect the catchment
evolution?

• *The reviewer is correct, the term "total cumulative rainfall since significant activity" reflects the total*
*rainfall since the end of Phase 5. This parameter is used as a proxy for catchment evolution within the*
*model under the assumption that in the absence of further eruptive activity hydrogeomorphic drainage*
*basin recovery will occur following the catchment disturbance associated with phase 5 (Pierson &*
*Major, 2014).*

Line 215-218. This point needs a better discussion in light of Figure 2 (see previous comment at line 124-129).

•   *As the reviewer identifies in their comment relating to line 124-129, large lahars are not exclusively*
*triggered in the wet season and there are examples of large lahars in the dry season. However, the*
*primary objective of the point in lines 215-218 is to emphasise that large lahars are frequently associated*
*with the passage of large synoptic weather systems which produce large volumes of total rainfall. The*
*increased frequency of rainfall events in the wet season (including such synoptic systems) results in an*
*increase in the average antecedent rainfall, which is identified as contributing to the observed reduction*
*in 1hr PRI based lahar initiation thresholds during the wet season.*

Line 225-227. This is questionable based on data here presented; see previous comment about figure 2.

•   *As identified by the reviewer, the term "absence of large lahars in the dry season" should be replaced*
*with "the reduction in the frequency of large lahars in the dry season" as there are a couple of examples*
*of such flows within the studied dataset. However, this reduction is still attributed to a combination of*
*the occurrence of fewer sustained catchment-wide synoptic weather systems and a reduction in average*
*antecedent rainfall and thus saturation level of pyroclastic deposits.*

- Lines 123-130: A description of the observation/detection methods used to identify lahars has been added in addition to information regarding how the lahars are categorised by magnitude. This is in response to Capra comment #1.

- Lines 131-143: Detail has been added regarding the use and definition of 1 hour peak rainfall intensity as discussed in Capra comments #4 and #5. This temporal resolution of rainfall data was the highest available in this case.

- Lines 133-137: A new figure has been created demonstrating timelines of rainfall data and lahar occurrence in response to Capra comment #3 and Pierson comment #17.

- Lines 144-154: Information regarding the methods used in the study (specifically analysis methods) has been transferred to this new consolidated methods section (Pierson comment #1). Some of this material has been moved to this section from later in the manuscript.

- Lines 173-175: A demonstration of the % of false positives present above an example threshold and details regarding the maximum non-lahar triggering rainfall intensity has been added. (Pierson comment #15).

- Lines 207-208: Clarity regarding what is meant by the term "cumulative rainfall since significant eruptive activity" has been added to address Capra comment #10.

- Lines 210-214: Information regarding the antecedent rainfall timescales used in other studies and the reasons for the different timescales has been added to address Capra comment #9 and Pierson comment #11.

- Line 242: Results of ROC analysis added as requested in Pierson comment #10.

- Lines 252-254: Pierson comment #8 has been addressed by adding information regarding the difficultly in making direct comparisons to the results of previous studies. Lines 263-264 also address this point by referencing a previous study which highlights an increase in lahar initiation thresholds with time.

- Line 277: Adjustment made to the phrasing as identified by Capra comment #12.

- Lines 279-280: References added to support point as suggested in Pierson comment #13.

- Lines 296-302: Information regarding the rainfall timescales used in previous studies of lahar initiation thresholds has been added (Pierson comment #6, Capra comment #4).

- Line 335: The caption has been amended to add clarity to the figure as identified in Pierson comment #16

- Lines 337-339: New caption for new figure 3.

- Throughout the references section additional references have been added where appropriate.

•   Line 474: New Figure 3 (Capra Comment #3, Pierson Comment #17).

•   Line 485 Onwards: A new appendix has been created (including two tables and a figure) to describe
ROC analysis more fully as identified by Pierson comment #14.

[revised manuscript text omitted]

Fig.2

[Figure]

Fig. 3

**Commented [A24]:** New figure as suggested by Capra comment #3 and Pierson comment #17

[Figure]

Fig. 4

[Figure]

Fig. 5

[Figure]

[Figure]

Fig. 7

[Figure]

Commented [A25]: New appendix explaining ROC analysis in response to Pierson comment #14

**Appendix I**

Receiver Operating Characteristic (ROC) analysis is a statistical technique that is used to illustrate the diagnostic ability of a binary classifier system (i.e. a system that subdivides the elements of a given dataset into two groups, for example the presence or absence of a disease, a pass or a fail in a test etc.). The method was first developed by electrical and radar engineers during World War II, and has since been used in psychology, medicine, meteorology, and forecasting of natural hazards.

A graphical plot, or Receiver Operating Characteristics curve (ROC curve) is often used to illustrate the effect of varying the value of the classifying parameter (for example the number of cancer cells per microlitre of blood or the pass mark in the previous example). The ROC curve is generated by plotting the true positive rate (TPR) against the false positive rate (FPR) as the value of the classifying, or threshold parameter, is changed. There are four possible outcomes from a binary classifier (Table A1): (i) correct prediction of an event that really did occur = true positive; (ii) incorrect prediction of an event that did not occur = false positive; (iii) predicting no event when an event does happen = false negative; and (iv) correct prediction that no event occurs and no event really does occur = true negative.

Imagine a situation where there are 200 patients undergoing a medical test, where alpha is some diagnostic threshold for having a medical condition. At a given value of alpha, the contingency table could resemble Table A2.

Here, the TPR is the number of true positives divided by the total number of predicted positives (both true and false), or $70/(70+30) = 0.70$

The FPR is the number of false positives divided by the total number of predicted negatives (both true and false), or $28/(28+72) = 0.28$

Thus for this value of alpha, the corresponding point would plot at (0.63, 0.28) on Figure A1 (the white square).

By systematically varying the value of the threshold parameter alpha, a whole series of 2x2 contingency tables would be generated, producing an array of points in ROC space and hence a curve (the dashed line).

A 100% rate of prediction (all true positives) would plot at (0, 1) on Figure A1 (the grey circle), whereas a 50% accurate rate of prediction (i.e. guessing the outcome of a coin toss) would plot at (0.5, 0.5). Random guesses thus plot along a diagonal line: points above the line represent predictions better than random, points below the line predictions worse than random.

**Appendix I: Table Captions**
**Table A1: 2x2 contingency table showing the possible outcomes of a binary classifier system.**
**Table A2: 2x2 contingency table for 200 patients undergoing a medical test for the presence or absence of**
**a condition.**
**Appendix I: Figure Captions**
**Fig. A1: ROC space and plots of the prediction examples discussed in the text.**

Table A1

| Total population | Event happens | Event does not happen |
| --- | --- | --- |
| Predict it happens | True positive | False positive |
| Predict it does not happen | False negative | True negative |

**Formatted Table**

Table A2

|  | Has condition | Has no condition |
|---|---|---|
| Predict has condition | 70 | 30 |
| Predict has no condition | 28 | 72 |

Formatted Table

Fig. A1

[Figure]

________________________________________________